# A Training-Free Framework for Long Video Understanding via Video-Query-Options Similarity

**Zhirong Wu**[1], **Xiaodong Wang**[1,2], **Langling Huang**[1], **Teng Xu**[3], **Peixi Peng**[1,2*]

[1]School of Electronic and Computer Engineering, Peking University,
[2]Pengcheng Laboratory,    [3]Douyin Group
{wuzhirong@stu., pxpeng@}pku.edu.cn

## Abstract

Multimodal Large Language Models (MLLMs) have achieved remarkable success in image and short video understanding tasks, but their performance on hour-long videos remains limited due to constraint of input token capacity. Existing approaches often require costly training procedures, hindering their adaptability to rapidly evolving MLLM architectures. In this paper, we propose a training-free framework for long video understanding, integrating three key innovations: Adaptive Frame Sampling (AFS), Dynamic Resolution Allocation (DRA), and Video-Query-Options Similarity (VQOS). AFS adaptively increases frame sampling density in highly relevant video segments to preserve critical temporal details, while DRA reduces spatial resolution in less relevant segments to suppress redundant information. VQOS enhances similarity calculation by prompting MLLMs to generate candidate answer options, fusing queries with options to refine relevance estimation. Mirroring human cognitive processes (hypothesis generation → focused verification → irrelevance filtering), our framework effectively improve model accuracy without fine-tuning. The method is implemented on LLaVA-Video and Qwen2.5-VL respectively, and experimental results show our method could achieve state-of-the-art performances over 5 mainstream benchmarks. Code is available in `https://github.com/wuzhirong520/VTR-VLM`.

## 1 Introduction

In recent years, Multimodal Large Language Models (MLLMs) have made rapid progress (Zhang et al., 2024b; Wang et al., 2024a; Liu et al., 2025b; Zhang et al., 2025a; Bai et al., 2025b; Zhu et al., 2025), with various post-training (Wang et al., 2025c; 2026; Bai et al., 2025a) and reasoning methods (Guo et al., 2025; Wang & Peng, 2025; Wang et al., 2025a) being developed to enhance video understanding. However, for long videos—especially hour-long videos—existing MLLMs generally exhibit limited capabilities, leaving substantial room for improvement.

The fundamental challenge in long video understanding stems from the inherent contradiction between limited model context windows and the vast spatiotemporal extent of video content. To mitigate this, some methods (Zhang et al., 2024a; Chen et al., 2024; Shen et al., 2025) extend the maximum token capacity through techniques such as parallel processing or multi-stage training. Others (Shen et al., 2024; Qin et al., 2025; Wang et al., 2025d; Li et al., 2024) exploit the inherent temporal and spatial redundancy in videos, aiming to compress the input by retaining only the most informative tokens. However, these methods typically require extensive and costly training procedures. Recently, training-free approaches (Ma et al., 2025b; Wang et al., 2025b; Tang et al., 2025) have demonstrated great potential by extracting meaningful and representative information from long videos without requiring model fine-tuning. In the field of training-free long video understanding, retrieval-based strategies offer a viable solution. Leveraging the strong performance of existing vision-language models of existing vision-language models (Radford et al., 2021; Zhai et al., 2023) in image-text retrieval, some methods (Tang et al., 2025; Liu et al., 2025a) effectively

---

*Corresponding author.

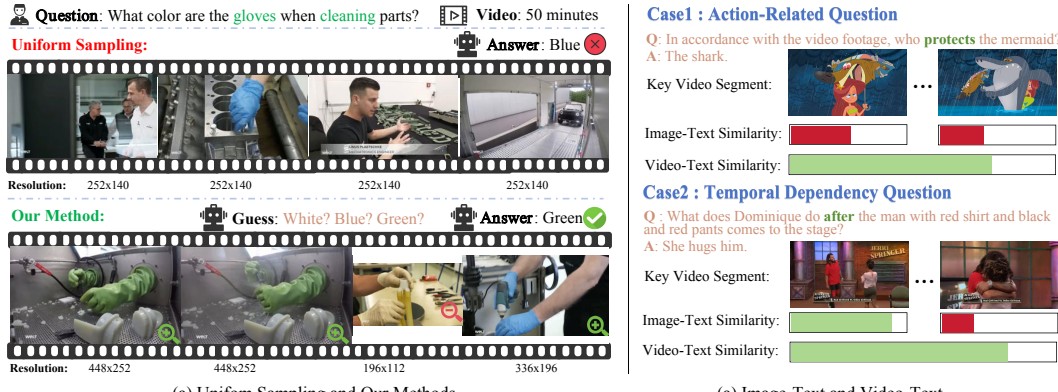

Figure 1: **Comparisons with existing methods.** (a) Most existing MLLMs rely on uniform frame sampling, which—due to context length limits—often misses critical information in long videos. Our approach enhances MLLMs' question-answering capabilities by retrieving question/options-relevant frames, densely sampling highly pertinent segments, and employing elevated resolution settings. (b) While some approaches leverage image-text retrieval, they are inadequate for action-related or temporally dependent questions, as static frames cannot reliably capture dynamic cues or temporal structure. Video-text retrieval is therefore essential in our method.

retrieve key frames that are semantically relevant to the user's query, thereby enabling MLLMs to focus on critical content. However, these approaches exhibit two key shortcomings: (1) they predominantly focus on static image, neglecting important information in videos such as actions, causal relationships and temporal dynamics, as shown in Fig. 1 (b), and (2) while effective at selecting key frames, they lack mechanisms to deeply exploit or refine the information contained in those frames.

In this paper, we conduct an in-depth exploration of such training-free retrieval-based methods for long video understanding. Building upon the state-of-the-art (SOTA) video-text retrieval model (Bolya et al., 2025), which computes similarity scores between video segments and textual queries, we introduce two key components: Adaptive Frame Sampling (AFS) and Dynamic Resolution Allocation (DRA). The former adaptively samples a greater number of frames from video segments with higher similarity scores, thereby enhancing the representational richness of relevant content. The latter reduces the spatial resolution of less relevant segments, optimizing both computational efficiency and focus on salient regions. To further refine the similarity computation, we propose Video-Query-Options Similarity (VQOS), a novel strategy wherein original MLLM is prompted to generate plausible answer options based on the user query. The similarity between the video and each generated option is then computed and fused with the original query for a more robust relevance estimation. Our approach closely emulates the cognitive process by which humans comprehend and answer questions about long videos: when faced with a question, humans typically formulate several hypotheses, selectively review the video to verify these hypotheses, and naturally filter out irrelevant information during the process. By mimicking this behavior, our method effectively improves accuracy in long video understanding without requiring any model fine-tuning. A typical example is shown in Fig. 1 (a) to illustrate our method.

We integrate our method into MLLMs including LLaVA-Video (Zhang et al., 2024b) and Qwen2.5-VL (Bai et al., 2025b), across both 7B and 72B parameter scales. Experimental results on 5 long video understanding benchmarks demonstrate an average performance gain of 5.3% and 5.0% in 7B size and 3.6% and 3.2% in 72B size compared with LLaVA-Video and Qwen2.5-VL. Especially, on hour-long video benchmarks LVBench (Wang et al., 2024b) and VideoEval-Pro (Ma et al., 2025a), our 7B-scale model achieves substantial improvements, outperforming LLaVA-Video and Qwen2.5-VL by an average of 8.5% and 8.3%, respectively. Our contributions are summarized as follows:

- We propose AFS and DRA to adaptively optimize frame selection and resolution resizing based on relevance scores for long video understanding.

- We develop VQOS mechanism, which leverages the capabilities of MLLMs to generate candidate answers and enhance similarity estimation through multi-hypothesis fusion.

- We integrate our method into LLaVA-Video and Qwen2.5-VL across 7B and 72B scales, achieving significant performance gains on 5 long video benchmarks—demonstrating its effectiveness and scalability in long video understanding.

## 2 RELATED WORK

### 2.1 MLLMs FOR LONG VIDEO

Existing approaches address the long video understanding challenge through two primary strategies: context extension (Zhang et al., 2024a; Chen et al., 2024; Shen et al., 2025) and token compression (Shen et al., 2024; Qin et al., 2025; Wang et al., 2025d; Li et al., 2024). The context extension strategy focuses on increasing the maximum sequence length that models can process during training. For example, LongVILA (Chen et al., 2024) employs multi-stage training pipelines and novel parallelism techniques to expand contextual capacity. In contrast, token compression methods aim to preserve more informative content within a reduced number of tokens. InternVideo2.5 (Wang et al., 2025d) adopts hierarchical token compression combined with task-preference optimization to improve representation efficiency, while VideoXL-2 (Qin et al., 2025) introduces task-aware key-value (KV) sparsification to enhance memory utilization. However, these methods require expensive training, limiting their adaptability to rapidly evolving MLLM architectures.

### 2.2 TRAINING-FREE LONG VIDEO UNDERSTANDING

Training-free approaches for long video understanding aim to extract meaningful and representative information without requiring model fine-tuning. These methods can be broadly categorized into three types: (1) Agent-based approaches (Zhang et al., 2023; Wang et al., 2024c; Luo et al., 2024; Ma et al., 2025b; Pang & Wang, 2025; Zhang et al., 2025b) involves dividing a long video into shorter clips, where agents generate descriptive captions for each clip and subsequently use these textual summaries to answer questions. For instance, Deep Video Discovery (Zhang et al., 2025b) utilizes LLM-based agents to autonomously explore and reason over segmented clips. In contrast to our method, these approaches are fundamentally dependent on video captioning, a process that is computationally intensive and prone to loss of fine-grained visual details due to the abstraction of rich visual content into text. Furthermore, they often rely on proprietary models such as GPT-4o (Hurst et al., 2024) as the underlying agent, making direct comparison unfair and impractical. (2) Compression-based approaches (Wang et al., 2025b; Luo et al., 2025; Gao et al., 2025) focus on reducing redundancy in the visual token stream, enabling more efficient processing of long video sequences. These methods typically operate on the internal token representations within the MLLM, compressing or pruning less informative visual tokens. For instance, AdaReTake (Wang et al., 2025b) adaptively removes redundant information in the key-value (KV) cache across both temporal and layer dimensions, allowing MLLMs to process up to 2048 frames efficiently. These techniques are complementary to our approach: while they operate at the token level during inference, our method targets the input frame-sampling stage prior to model ingestion. (3) Retrieval-based approaches (Park et al., 2024; Tang et al., 2025; Liu et al., 2025a) employ lightweight expert models to identify and retrieve keyframes that are most relevant to the given question. For instance, AKS (Tang et al., 2025) computes frame-question similarity using CLIP (Radford et al., 2021) embeddings to select semantically aligned frames. Our method is closely aligned with this paradigm; however, we introduce a more refined similarity estimation strategy by leveraging a dedicated video-text retrieval model to jointly compute the similarity among the video, the question, and the candidate options. Furthermore, we utilize the resulting similarity scores not only for adaptive frame sampling but also for dynamic frame resizing, thereby enhancing the quality of the input representation.

### 2.3 VIDEO-TEXT RETRIEVAL

Video-Text Retrieval (VTR) is a cross-modal task that aims to measure the semantic similarity between video content and textual queries. Vision language models like CLIP (Radford et al., 2021) and SigLIP (Zhai et al., 2023) excell at image-text retrieval, and can also zero-shot to video-text retrieval by pooling image embdeddings. CLIP4CLIP (Luo et al., 2022) fine-tunes the CLIP model for video-text retrieval leavaging contrastive learning. Recently, PerceptionEncoder (Bolya et al., 2025)

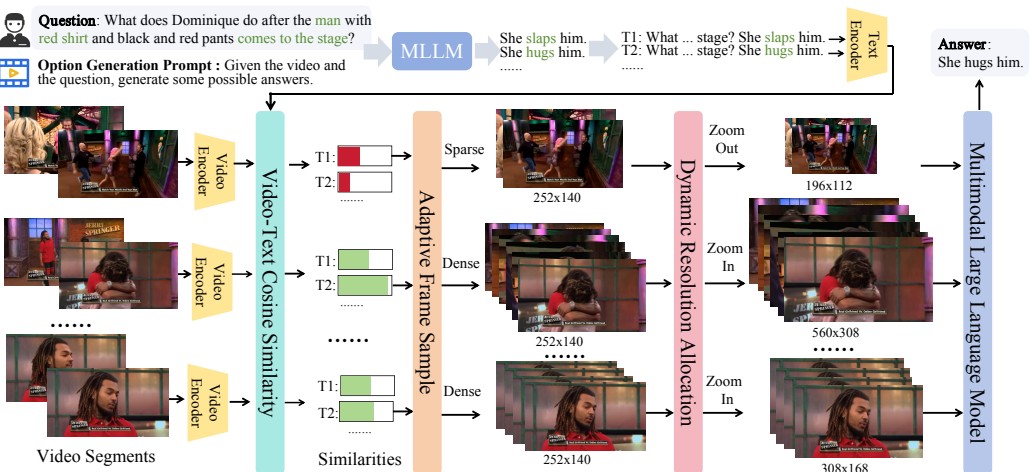

Figure 2: **Overall Framework**. We first generate plausible answer options using the original MLLM, concatenate them with the question, and compute similarity scores between the resulting queries and video segments using a pre-trained video-text retrieval model. Based on these similarity scores, Adaptive Frame Sampling increases frame density in high-similarity regions, while Dynamic Resolution Allocation increases resolution in more relevant segments.

pretrained visual encoder with extensive video data, which makes it have excellent performance in video text retrieval task.

## 3 METHODOLOGY

The overall framework of our approach is illustrated in Fig. 2. By generating options and leveraging a video-text retrieval model to compute their similarity with video segments (Sec. 3.1), we guide both adaptive frame sampling (Sec. 3.2) and dynamic resolution allocation (Sec. 3.3). Specifically, regions with higher similarity receive denser frame sampling and higher spatial resolution. For clarity, we provide pseudocode in Sec. 3.4.

### 3.1 VIDEO-QUERY-OPTIONS SIMILARITY

Given a long video, we uniformly divide it into $m$ equal-length segments, denoted as :

$$\mathcal{V} = \{V_1, V_2, \ldots, V_m\}. \tag{1}$$

For each segment $V_i$ and user query $Q$, we employ a video-text retrieval (VTR) model to extract the video and text features, denoted as $f_{v_i}, f_q \in \mathbb{R}^d$. The initial similarity score $S_i^0$ for the $i$-th video segment is then computed as the cosine similarity between the video and text features:

$$S_i^0 = \frac{f_{v_i} \cdot f_q}{\|f_{v_i}\| \cdot \|f_q\|}. \tag{2}$$

Then we prompt the MLLM to generate $z$ candidate options, where the prompt is shown in Fig. 2. Subsequently, the question is concatenated with each option to form $z$ distinct statements:

$$\mathcal{T} = \{T_1, T_2, ...T_z\}. \tag{3}$$

These statements are encoded into text features using the same VTR model respectively, and combined as $\mathcal{F}$. The final similarity score $S_i$ to for the $i$-th video segment is then computed as the maximum cosine similarity between the video feature and all text features in $\mathcal{F}$:

$$S_i = \max_{f \in \mathcal{F}} \frac{f_{v_i} \cdot f}{\|f_{v_i}\| \cdot \|f\|}. \tag{4}$$

Note that $S_i^0$ is utilized for selecting video segments based on user queries without options, and $S_i$ is employed when the options are available. Both undergo the same processing pipeline described in Sec. 3.2 and Sec. 3.3. To enhance diversity in option generation, options can be generated multiple rounds by splitting the whole video into several parts and generating options separately. Specifically, for multiple-choice questions, the option generation can be bypassed, as candidate answers are provided.

## 3.2 ADAPTIVE FRAME SAMPLING

To effectively represent long video with a limited number of frames, we adopt a adaptive sampling strategy that emphasizes semantically relevant content. Instead of treating all video segments equally, we leverage their similarity to the input query to guide the sampling process.

To sample $N$ frames from a video, we first select the top-$k$ segments based on their video-query-options similarity scores $S_1, S_2, \ldots, S_k$. From each selected segment $V_i$, we uniformly sample $p_i$ frames such that the following constraints are satisfied:

$$\sum_{i=1}^{k} p_i = N, \quad S_i \leq S_j \Rightarrow p_i \leq p_j. \tag{5}$$

These conditions ensure that the total number of sampled frames remains fixed at $N$, and segments with higher similarity scores are assigned more frames, while segments with lower scores are assigned fewer.

To simplify the allocation process while adhering to the desired priority, we first sort the top-$k$ segments in descending order of their similarity scores. These sorted segments are then partitioned into $L_1$ sampling levels, where the $l$-th level contains $m_l$ segments, each segment samples $c_l$ frames , satisfying that:

$$\sum_{l=1}^{L_1} m_l = k, \quad \sum_{l=1}^{L_1} m_l \cdot c_l = N, \tag{6}$$

where $c_1 > c_2 > ... > c_{L_1}$. For simplicity, we assume roughly equal segment distribution across levels, utilizing Pulp (Mitchell et al., 2011) to find a feasible $\{m_l\}$ with predefined $\{c_l\}$.

This sampling strategy allows more relevant segments to contribute a greater number of frames, thereby enhancing the semantic coverage and representativeness of the selected frame set while maintaining computational simplicity.

## 3.3 DYNAMIC RESOLUTION ALLOCATION

For long video sequences, existing visual encoders often encounter a fundamental trade-off between the number of processed frames and spatial resolution: given a fixed total input token budget (determined by the model's sequence length limit), increasing the number of frames necessitates downscaling the spatial resolution of frames, which may result in the loss of critical spatial information in high-importance frames. To mitigate this issue, we propose an Dynamic Resolution Allocation strategy that allocates higher resolution to key frames (frames of high task relevance) and lower resolution to non-key frames.

For a given video with resolution $H \times W$ , we define $L_2$ resolution levels as:

$$H_i = \left\lfloor \alpha_i \frac{H}{I} \right\rfloor \cdot I, \quad W_i = \left\lfloor \alpha_i \frac{W}{I} \right\rfloor \cdot I, \tag{7}$$

where $\alpha_1 > \alpha_2 > ... > \alpha_{L_2} \in (0, 1]$ are scaling factors that preserve the aspect ratio, and $I = 28$ is a stride constraint imposed by the visual encoder architecture due to patch size and downsampling requirements. Each resolution $(H_i, W_i)$ is further constrained to lie within a valid range $C_{min} \leq H_i, W_i \leq C_{max}$.

A feasible resolution allocation strategy $\{n_1, n_2, ..., n_L\}$ must satisfy two constraints: (1) All frames are allocated to $L$ resolution levels, (2) The sum of tokens across all frames equals the budget $P$:

$$\sum_{i=1}^{L} n_i = N, \quad \sum_{i=1}^{L} (n_i \cdot H_i \cdot W_i) = P. \tag{8}$$

Frames with higher similarity scores are assigned to higher-resolution levels, for example $n_1$ frames with largest similarity scores are allocated resolution $(H_1, W_1)$.

To simplify computation while maintaining a balanced distribution of tokens across resolution levels, we assume a uniform per-level token budget — that is, each resolution level receives an equal share of the total token budget P . Under this assumption, the number of frames allocated to resolution level $i$ can be approximated by:

$$\hat{n}_i = \left\lfloor \frac{P}{L \cdot H_i \cdot W_i} \right\rfloor. \tag{9}$$

### 3.4 Pseudocode of Our Method

As outlined in Algorithm 1, given a set of pre-segmented video clips $\mathcal{V}$ and a question $q$ , our method first encodes the question and all video segments using a VTR model to obtain embeddings. Initial video–question similarity scores ($S^0$) are computed via cosine similarity, and the top-$R \cdot N$ most relevant segments are selected to form a refined candidate pool $\mathcal{V}'$. Over $R$ rounds of option generation, the algorithm samples $N$ segments from $\mathcal{V}'$ per round, applies Adaptive Frame Sampling (AFS) guided by $S^0$, and enhances frame quality via Dynamic Resolution Allocation (DRA). These processed frames are fed into an MLLM to generate diverse candidate answer options. Each option is then combined with the original question, re-encoded by the VTR model, and scored against all segment embeddings; the maximum similarity across options yields a refined relevance score. Finally, the top-$N$ segments are retrieved based on this similarity score, resampled with AFS and DRA, and passed to the MLLM together with the question to produce the final answer. Note that the $R$ rounds of option generation are not temporally dependent and can be parallelized for acceleration.

---

**Algorithm 1** Our Method

---

**Require:** Video segments ($\mathcal{V}$), question ($q$), option generation round ($R$), sampled frame num ($N$), video-text retrieval model (VTR), multimodal large language model (MLLM), Adaptive frame sampling (AFS), Dynamic Resolution Allocation (DRA).
**Ensure:** Answer the question according to the video.
1: $segment\_embeddings \leftarrow \emptyset$
2: $simiarities \leftarrow \emptyset$
3: $generated\_options \leftarrow \emptyset$
4: $q\_embedding \leftarrow \text{VTR}(q)$
5: **for** each $V_i \in \mathcal{V}$ **do**
6:    $segment\_embedding.\text{append}(\text{VTR}(V_i))$
7: **end for**
8: $S^0 \leftarrow cosine\_simiarity(q\_embedding, segment\_embeddings)$
9: $\mathcal{V}' \leftarrow TopK(\mathcal{V}, S^0, R \cdot N)$
10: **for** $r \leftarrow 1$ to $R$ **do**
11:    $\mathcal{V}_r \leftarrow (V_{r+kR})_{k=0}^{N-1}, \quad V \in \mathcal{V}'$
12:    $sampled\_frames \leftarrow \text{DRA}(\text{AFS}(S^0, \mathcal{V}_r))$
13:    $options \leftarrow \text{MLLM}('Please \quad generate \quad some \quad options....', sampled\_frames)$
14:    $generated\_options.\text{extend}(options)$
15: **end for**
16: **for** each $o \in generated\_options$ **do**
17:    $o\_embedding \leftarrow \text{VTR}(q + o)$
18:    $S \leftarrow cosine\_simiarity(o\_embedding, segment\_embeddings)$
19:    $similarities \leftarrow \max(simiarities, S)$
20: **end for**
21: $\mathcal{V}_{final} \leftarrow TopK(\mathcal{V}, simiarities, N)$
22: $sampled\_frames \leftarrow \text{DRA}(\text{AFS}(simiarities, \mathcal{V}_{final}))$
23: $answer \leftarrow \text{MLLM}(q, sampled\_frames)$
24: **return** $answer$

---

Table 1: Comparisons on widely used benchmarks. **LongVB** and **VMME** refer to LongVideoBench and VideoMME, respectively.

| Model | Size | LVBench (67min) | MLVU (13min) | LongVB (8min) | VMME (17min) | VideoEval-Pro (38min) | | Average |
|---|---|---|---|---|---|---|---|---|
| | | Overall | M-Avg | Overall | Overall | Open | MCQ | |
| **Proprietary MLLMs** | | | | | | | | |
| Gemini-1.5-Pro | - | 33.1 | - | 64.0 | 75.0 | 39.3 | 63.4 | - |
| GPT-4o | - | 48.9 | 64.6 | 66.7 | 71.9 | 34.2 | 59.5 | 57.6 |
| Seed1.5-VL | 200B | 64.6 | 82.1 | 74.0 | 77.9 | 40.7 | 66.6 | 67.7 |
| **Open-Source MLLMs** | | | | | | | | |
| Qwen2-VL | 7B | 44.2 | 69.8 | 55.6 | 63.3 | 26.5 | 48.2 | 51.3 |
| NVILA | 8B | 42.6 | 70.6 | 57.7 | 64.2 | - | - | - |
| VideoLLaMA3 | 7B | 45.3 | 73.0 | 59.8 | 66.2 | - | - | - |
| InternVL3 | 8B | - | 71.4 | 58.8 | 66.3 | 24.7 | 48.4 | - |
| **Training-Based MLLMs For Long Video** | | | | | | | | |
| LongVA | 7B | - | 56.3 | - | 52.6 | 16.5 | 38.0 | - |
| VideoChat-Flash | 7B | 48.2 | 74.7 | 64.7 | 65.3 | 27.0 | 51.2 | 55.2 |
| InternVideo2.5 | 8B | 46.4 | 74.9 | 60.6 | 65.1 | 27.2 | 53.2 | 54.6 |
| Video-XL-2 | 8B | 48.4 | 74.8 | 61.0 | 66.6 | 28.6 | 53.0 | 55.4 |
| **Training-Free MLLMs For Long Video** | | | | | | | | |
| LLaVA-Video | 7B | 42.0 | 69.3 | 57.4 | 63.2 | 24.2 | 47.6 | 50.6 |
| + AKS | | 47.0 | 69.1 | 62.9 | 65.3 | 28.9 | 51.3 | 54.1 |
| + AdaReTake | | 49.6 | 70.6 | 59.6 | 64.0 | 27.7 | 53.5 | 54.2 |
| + Ours-GO | | 51.3 | 70.3 | 61.0 | 64.8 | 32.7 | 55.3 | 55.9 |
| + Ours-PO | | 54.2 | 73.4 | 61.0 | 65.5 | - | 56.9 | 57.3 |
| Qwen2.5-VL | 7B | 45.5 | 69.4 | 61.0 | 66.4 | 27.7 | 46.6 | 52.8 |
| + AdaReTake | | 51.0 | 72.9 | 61.9 | 67.4 | 30.8 | 53.7 | 56.3 |
| + Ours-GO | | 52.7 | 72.3 | 63.4 | 66.7 | 35.0 | 56.9 | 57.8 |
| + Ours-GO + AdaReTake | | 55.5 | 74.3 | 63.0 | 69.3 | 35.4 | 57.3 | 59.1 |
| + Ours-PO | | 55.5 | 74.1 | 63.3 | 67.9 | - | 57.6 | 58.9 |
| + Ours-PO + AdaReTake | | 57.5 | 74.7 | 64.2 | 69.4 | - | 58.2 | 59.9 |
| LLaVA-Video | 72B | 46.1 | 71.3 | 62.4 | 70.3 | 26.7 | 50.1 | 54.5 |
| + Ours-GO | | 51.7 | 71.7 | 63.6 | 70.3 | 33.1 | 58.3 | 58.1 |
| + Ours-PO | | 54.8 | 74.7 | 64.0 | 70.3 | - | 60.1 | 59.5 |
| Qwen2.5-VL | 72B | 49.6 | 75.3 | 65.1 | 73.3 | 29.9 | 55.9 | 58.2 |
| + Ours-GO | | 54.0 | 76.9 | 66.3 | 72.7 | 36.5 | 61.9 | 61.4 |
| + Ours-PO | | 56.9 | 77.7 | 66.3 | 73.1 | - | 64.2 | 62.5 |

Table 2: Ablation for components. For LongVideoBench, VideoMME, and VideoEval-Pro, we evaluate on representative and cost-efficient subsets: **LVB-L** and **VMME-L** (the long-video subsets of LongVideoBench and VideoMME, respectively) and **VEP-M** (the multiple-choice subset of VideoEval-Pro).

| Method | LVBench | MLVU | LVB-L | VMME-L | VEP-M | $\Delta_{avg}$ |
|---|---|---|---|---|---|---|
| **LLaVA-Video-7B** | 42.0 | 69.3 | 48.2 | 51.4 | 47.6 | - |
| + top-$N$ frames retrieval | 49.6 | 70.0 | 55.3 | 51.8 | 53.5 | +4.3 |
| + top-$k$ segments uniform sampling | 48.9 | 70.3 | 55.7 | 53.2 | 53.1 | +0.2 |
| + adaptive frame sampling | 50.2 | 70.4 | 56.6 | 53.2 | 54.2 | +0.7 |
| + generated options | 51.3 | 70.3 | 57.1 | 54.0 | 55.3 | +0.7 |
| + provided options | 54.2 | 73.4 | 55.9 | 55.0 | 56.9 | +1.5 |
| **Qwen2.5-VL-7B** | 45.5 | 69.4 | 53.7 | 55.6 | 46.6 | - |
| + top-$N$ frames retrieval | 50.3 | 70.0 | 53.5 | 55.4 | 51.4 | +2.0 |
| + top-$k$ segments uniform sampling | 50.1 | 70.3 | 53.9 | 55.1 | 51.1 | +0.0 |
| + adaptive frame sampling | 51.1 | 70.0 | 55.3 | 55.7 | 52.7 | +0.8 |
| + dynamic resolution allocation | 51.9 | 72.1 | 59.0 | 56.0 | 55.9 | +2.1 |
| + generated options | 52.7 | 72.3 | 58.5 | 56.4 | 56.9 | +0.4 |
| + provided options | 55.5 | 74.1 | 58.9 | 56.9 | 57.6 | +1.2 |

# 4 EXPERIMENTS

## 4.1 IMPLEMENTATION DETAILS

We divide videos into 16-second segments and employ PE-G/14 (Bolya et al., 2025) for video-text retrieval with $fps = 1$. Then we integrate our method into both the 7B and 72B variants of LLaVA-Video (Zhang et al., 2024b) and Qwen2.5-VL (Bai et al., 2025b). For LLaVA-Video, we limit the input to a maximum of 64 frames, select top-16 segments, and set the frame sampling level to $\{2, 4, 8\}$ without employing DRA, as it is designed to accept fixed-resolution inputs of $384 \times 384$. For Qwen2.5-VL, we set the maximum number of frames to 768, select top-48 segments, set the frame sampling level to $\{8, 16, 32\}$, and constrain the resolution budget to $20480 \times 28 \times 28$ with resolution levels ranging from 84 to 644. Furthermore, we rerun two typical training-free methods AKS (Tang et al., 2025) and AdaReTake (Wang et al., 2025b) on the datasets which are not reported in the original papers, and integrate AdaReTake into our method using 2048 frames as input.

Two versions of our method are implemented: (1) **Ours-GO** which means the candidate options are generated by original MLLM without additional prior knowledge, and (2) **Ours-PO** which indicates the candidate options are given by the dataset. Empirically, we generate options three iterations for LLaVA-Video and once for Qwen2.5-VL. Ours-GO is more general and the comparisons with other methods are fair. In contrast, Ours-PO only adapts to the multiple-choice questions, and the results are just for reference.

## 4.2 BENCHMARKS

Five widely used datasets are used, including: (1) **LVBench** (Wang et al., 2024b) is a benchmark designed for evaluating extreme long video understanding. (2) **MLVU** (Zhou et al., 2025) is a multi-task benchmark for long video understanding. We report the M-Avg metric on the *dev* set. (3) **LongVideoBench** (Wu et al., 2024) is a benchmark for both short and long video understanding. We report the overall accuracy on its *val* set without interleaved subtitles. (4) **VideoMME** (Fu et al., 2025) is the first-ever comprehensive video understanding benchmark. We report the accuracy results without using subtitles. (5) **VideoEval-Pro** (Ma et al., 2025a) is a benchmark designed for more robust evaluation of long video understanding capabilities. It consists of 465 videos, with each video exceeding 10 minutes, selected from the four aforementioned benchmarks. Unlike previous benchmarks that primarily rely on multiple-choice questions — which may allow models to exploit answer options through guessing — VideoEval-Pro adopts open-ended questions, offering a more robust, comprehensive, and realistic assessment of models' long video comprehension abilities. The open-ended and multiple-choice metrics are reported. Overall, the average video lengths of these benchmarks are about 67 minutes, 13 minutes, 8 minutes, 17 minutes and 38 minutes, respectively. More evaluation details could be found in Appendix D.

## 4.3 MAIN RESULTS

The compared results are shown in Table 1. It is evident that:

(1) Compared with LLaVA-Video and Qwen2.5-VL across two model scales (7B and 72B), Ours-Go demonstrates substantial performance gains on five prominent long-form video benchmarks. Specifically, the 7B-scale models exhibit performance improvements of 5.3% and 5.0% respectively, and the 72B-scale models could achieve gains of 3.6% and 3.2% respectively. It shows our method could improve the baseline significantly. Ours-PO outperforms Ours-GO in most cases due to the guaranteed presence of correct answers in provided options. It indicates the method could leverage the prior knowledge of the options.

(2) Ours-Go outperforms the SOTA training-free approaches AKS and AdaReTake overall. Ours-Go achieves comparable performances in datasets with relatively shorter videos including MLVU, LongVideoBench and VideoMME, while shows clear advantage in longer video datasets LVBench and VideoEval-Pro (more detailed results on these two benchmarks are in Appendix B). Specifically, based on LLaVA-Video-7B, Ours-Go outperforms AdaReTake by an average of 0.6% on the first three datasets and by 2.8% on the latter two. Compared with AKS, Ours-Go lags slightly by an average of 0.4% on the former three datasets but surpasses it by an average of 3.0% on the latter two. It shows our potential for long video understanding.

(3) Since AdaReTake operates via token-level compression, we could further integrate the two approaches, yielding improved results of 1.3% and 1.0% as shown in "Ours-Go + AdaReTake" and "Ours-PO + AdaReTake".

(4) Overall, our method achieves SOTA average performance except proprietary models on these benchmarks without any training. This underscores the effectiveness and practicality of our approach in real-world long video understanding scenarios.

Since AKS doesn't provide code for Qwen2.5-VL, here we compare it only on LLaVA-Video for rigorous. We don't compare with other training-free methods due to differing used models and missing results on certain benchmarks, making direct comparison infeasible. More detailed comparisons with AKS and results of other training-free methods are included in Appendix A for reference.

## 4.4 ABLATION STUDIES

**Effect of components.** We demonstrate performance improvements after integrating partial components of our method into LLaVA-Video-7B and Qwen2.5-VL-7B, as shown in Table 2. We could obtain the follow key findings: (1) Image-text retrieval for top-$N$ frames yields significant gains of 4.3% and 2.0% respectively. Retrieving top-$k$ video segments with uniform sampling within each segment provides only marginal gains of 0.2% and 0.0%. It indicates that direct retrieval on video segments plays similar role with frame retrieval. (2) In contrast, incorporating our AFS based on video-text retrieval provides additional improvements of 0.7% and 0.8%, and our DRA in Qwen2.5-VL-7B contributes a substantial 2.1% gain. The reason is that the proposed AFS and DRA could focus on more important cues adaptively. (3) Furthermore, utiliz-

Table 3: Ablation for VTR models on LVBench. **Res.** refers to video resolution. **Time** refers to average inference time (second) per minute video.

| Model | Size | Res. | Acc | Time |
|---|---|---|---|---|
| CLIP-B/32 | 0.2B | 224 | 49.8 | 0.6 |
| CLIP-B/16 | 0.2B | 224 | 50.5 | 0.7 |
| CLIP-L/14 | 0.4B | 224 | 50.8 | 0.9 |
| CLIP-L/14-336 | 0.4B | 336 | 51.2 | 1.7 |
| PE-L/14 | 0.6B | 336 | 52.0 | 1.8 |
| PE-G/14 | 2.4B | 448 | 54.2 | 15.5 |
| SigLIP-LLaVA | 0.8B | 384 | 52.0 | - |

ing our generated options for similarity computation yields 0.7% and 0.4% improvements, while leveraging provided multiple-choice options achieves further gains of 1.5% and 1.2%.

**Ablation for VTR models.** Table 3 presents zero-shot video-text retrieval results using image-text retrieval models (CLIP) and pretrained video-text retrieval models (PerceptionEncoder). Results demonstrate that models with larger parameters, higher resolution, and video-specific pre-training could enhance performance. For computation constrained scenarios, smaller models remain viable alternatives (Appendix G.1). We also introduce "SigLIP-LLaVA", which directly reuses LLaVA-Video's visual encoder—requiring no additional VTR model (see Appendix G.2). Moreover, in streaming scenarios, VTR inference time can be effectively hidden by overlapping it with video ingestion latency, rendering it negligible (see Appendix G.3).

**Ablation for segment length and sample fps.** Table 4 shows results for LLaVA-Video-7B and Qwen2.5-VL-7B on LVBench. When segment length is 0, video segments will degenerate to single images. Results indicate that segment lengths of 8s or 16s yield superior performance, with VTR model sampling fps of 1.0 being optimal. When the segment length is too long (e.g., 32s or 64s), performance drops due to loss of fine-grained event details. Lower fps and longer segments reduce computational cost, making this configuration suitable for resource-constrained scenarios.

Table 4: Ablation for segment length and fps.

| Model | FPS | Segment Length | | | | |
|---|---|---|---|---|---|---|
| | | 0 | 8s | 16s | 32s | 64s |
| LLaVA | 0.5 | 51.8 | 53.4 | 52.7 | 51.5 | 48.7 |
| | 1.0 | 51.4 | 54.2 | 52.0 | 50.8 | 47.4 |
| | 2.0 | 51.0 | 53.6 | 53.7 | 50.9 | 48.6 |
| Qwen | 0.5 | 48.9 | 54.3 | 54.5 | 54.0 | 50.3 |
| | 1.0 | 49.8 | 54.0 | 55.2 | 52.8 | 50.0 |
| | 2.0 | 49.2 | 54.8 | 54.3 | 52.9 | 49.8 |

**Ablation for video segment retrieval.** Table 5 shows results for LLaVA-Video-7B and Qwen2.5-VL-7B on LVBench with varying top-$k$ selections and different total frame counts. The re-

sults indicate that retrieving more video segments may degrade performance, due to the inclusion of irrelevant content and a reduction of information per segment. Additionally, to complement local retrieval, we also explore how global information can be integrated in Appendix F.

**Ablation for similarity computation.** Table 6 presents results on VideoEval-Pro and LVBench using LLaVA-Video-7B, comparing performance with question-only inputs, generated options, and provided options. We also report the upper bound of our method using correct answers for similarity computation. Results shows that provided options significantly enhance performance, while generated options are essential for open-ended questions lacking predefined options.

Table 5: Ablation for video segment retrieval.

| Model | Frames | Top-K | | | | |
|---|---|---|---|---|---|---|
| | | 4 | 8 | 16 | 32 | 64 |
| LLaVA | 16 | 48.6 | 47.7 | 46.0 | - | - |
| | 32 | 50.0 | 51.0 | 49.3 | 47.5 | - |
| | 64 | 51.9 | 53.3 | 54.2 | 48.5 | 46.7 |
| Qwen | 64 | 50.4 | 50.2 | 49.4 | 46.5 | 45.3 |
| | 256 | 51.6 | 51.5 | 52.2 | 52.6 | 49.8 |
| | 512 | - | 52.8 | 53.4 | 53.5 | 53.9 |
| | 1024 | - | - | 54.6 | 53.7 | 53.8 |

## 4.5 OPTION GENERATION QUALITY

To systematically evaluate the quality of the generated options, we introduce two complementary metrics: (1) **Option Coverage Accuracy** (OCA) — the proportion of questions for which at least one generated option is semantically equivalent to the ground-truth answer. A high OCA indicates that the generator reliably includes the correct answer within its output for most questions. (2) **Mean Proportion of Correct Options** (MPCO) — the average fraction of semantically correct options among all generated options. This metric penalizes models that generate many incorrect or irrelevant options alongside the correct one. Details of these metrics could be found in Appendix E.1.

Fig. 3 illustrates how the number of option generation rounds affects LLaVA-Video's performance on VideoEval-Pro in terms of answer accuracy, OCA, and MPCO. As the number of rounds increases from 1 to 4, OCA steadily rises (54.6% → 63.8%), indicating improved coverage of semantically correct options. In contrast, MPCO consistently declines (26.8% → 20.0%), reflecting the dilution of correct options in an expanding pool of distractors. Notably, model accuracy initially benefits from improved OCA (rising from 54.2% to 55.3% within 3 rounds), but begins to slightly decline thereafter (55.3% → 55.1%) — suggesting that beyond a certain point, the inclusion of additional distractors outweighs the gains from better option coverage, ultimately harming answer accuracy. More experiments about option generation could be found in Appendix E.2 and Appendix E.3.

Table 6: Ablation for similairty computation.

| Method | VideoEval-Pro | | LVBench |
|---|---|---|---|
| | Open | MCQ | |
| only question | 31.0 | 54.2 | 50.2 |
| generate 1 round | 31.2 | 54.3 | 50.7 |
| generate 2 rounds | 32.1 | 54.3 | 50.8 |
| generate 3 rounds | 32.7 | 55.3 | 51.3 |
| generate 4 rounds | 31.9 | 55.1 | 50.4 |
| provided options | - | 56.9 | 54.2 |
| correct answer | 36.4 | 60.9 | 58.6 |

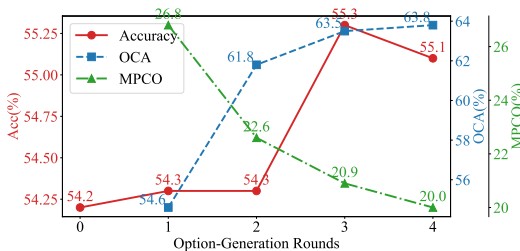

Figure 3: Impact of option generation rounds

## 5 CONCLUSION

We propose a training-free framework for long video understanding through dynamic adjustment of frame resolution and sampling density, leveraging similarities between video, query, and options. Evaluated on five benchmarks with multiple MLLMs, our approach achieves significant performance gains, demonstrating a scalable and effective paradigm for long-form video comprehension.

ACKNOWLEDGMENTS

The study was funded by the Shenzhen Science and Technology Program (KQTD20240729102051063), the National Natural Science Foundation of China under contracts No. 62422602, No. 62372010, No. 62425101, No. 62332002, No. 62372010, No. 62206281, and the major key project of the Peng Cheng Laboratory (PCL2021A13 and PCL2025A02). Computing support was provided by Pengcheng Cloudbrain.

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

Table 7: Comparisons on widely used benchmarks. **LongVB** and **VMME** refer to LongVideoBench and VideoMME, respectively.

| Model | LVBench | MLVU | LongVB | VMME | VideoEval-Pro | | Average |
|---|---|---|---|---|---|---|---|
| | Overall | M-Avg | Overall | Overall | Open | MCQ | |
| LLaVA-Video-7B | 42.0 | 69.3 | 57.4 | 63.2 | 24.2 | 47.6 | 50.6 |
| + AKS (CLIP) | 47.0 | 69.1 | 62.9 | 65.3 | 28.9 | 51.3 | 54.1 |
| + Ours-GO (CLIP) | 49.1 | 71.5 | 60.2 | 64.1 | 29.9 | 54.8 | 54.9 |
| + AKS (PE) | 46.4 | 69.1 | 60.7 | 64.1 | 30.0 | 52.9 | 53.9 |
| + Ours-GO (PE) | 51.3 | 70.3 | 61.0 | 64.8 | 32.7 | 55.2 | 55.9 |
| Qwen2.5-VL-7B | 45.5 | 69.4 | 61.0 | 66.4 | 27.7 | 46.6 | 52.8 |
| + AKS (CLIP) | 45.0 | 65.9 | 58.6 | 63.2 | 25.4 | 46.2 | 50.7 |
| + Ours-GO (CLIP) | 50.9 | 71.5 | 60.9 | 66.7 | 31.3 | 53.1 | 55.7 |
| + AKS (PE) | 46.1 | 67.1 | 58.8 | 62.4 | 26.0 | 47.6 | 51.3 |
| + Ours-GO (PE) | 52.7 | 72.3 | 63.4 | 66.7 | 35.0 | 56.9 | 57.8 |

# A  MORE QUANTITATIVE COMPARISON

## A.1  EXTRA COMPARISON WITH AKS

To fairly compare with AKS (Tang et al., 2025) under same retrieval model, two additional experiments are conducted: (1) Firstly, we replace the vision-language model in AKS with PE-G/14 (Bolya et al., 2025), as used in our method, while keeping all hyperparameters identical to those in the original AKS. This variant is denoted as "AKS (PE)". We compare it with the original AKS and "Ours-GO (PE)" (our method with PE-G/14); (2) The CLIP-B/32 (Radford et al., 2021) is used in our method ("Ours-GO (CLIP)"), which is consistent with the original AKS. We conduct experiments under both LLaVA-Video-7B (Zhang et al., 2024b) and Qwen2.5-VL-7B (Bai et al., 2025b) backbones. Since the AKS does not support Qwen2.5-VL-7B, we implement it with 768 frames to align with our method, while keeping all other hyperparameters unchanged. The results are presented in Table 7.

As shown in Table 7, when paired with LLaVA-Video-7B, AKS (PE) exhibits inferior performance compared to the original AKS using CLIP-B/32. Even though AKS (PE) outperforms AKS in combination with Qwen2.5-VL-7B, both fall short of the baseline Qwen2.5-VL-7B. Since AKS doesn't provide source code on PE-G/14 and Qwen2.5-VL-7B, these comparisons are not rigorous and just for reference. It indicates that the native employment of better retrieval model and backbone could not improve performances directly, which demonstrates our advantages.

Except for the generalization of hyperparameters, the reason may come from the method mechanism. AKS aims at maintaining temporal coverage across the video. This design is beneficial for relatively short videos (e.g., in LongVideoBench (Wu et al., 2024) and VideoMME (Fu et al., 2025)), it becomes less effective—and theoretically less meaningful—for hour-long videos, where the sheer duration and sparse distribution of critical events make coverage less important than precise, query-focused retrieval. In such scenarios, overemphasis on coverage may dilute the selection of highly relevant segments, ultimately harming performance. In contrast, our approach does not prioritize full-video coverage and instead focus on retrieval accuracy, leading to superior performance especially on longer video benchmarks (e.g., LVBench (Wang et al., 2024b) and VideoEval-Pro (Ma et al., 2025a)).

## A.2  OTHER TRAINING-FREE METHODS

More training-free methods are listed in Table 8: MRVideo (Pang & Wang, 2025), DeepDiscovery (Zhang et al., 2025b), MenVid (Yuan et al., 2025), QuoTA (Luo et al., 2025), E-VRAG (Xu et al., 2025) and APVR (Gao et al., 2025). However, due to the use of different (and in some cases unspecified) base models, as well as missing evaluation results on certain benchmarks, a direct comparison is not feasible; thus, the results are provided for reference only.

Table 8: More training-free methods. **LongVB** and **VMME** refer to LongVideoBench and VideoMME, respectively.

| Method | Used Models | LVBench | MLVU | LongVB | VMME |
|---|---|---|---|---|---|
| | | M-Avg | Overall | Overall | Overall |
| MRVideo | Gemini-2.0- Flash/GPT4o | 60.8 | - | - | - |
| DeepDiscovery | GPT-4.1+OpenAI o3 | 74.2 | - | 71.6 | - |
| MemVid | LanguageBind-Large+Qwen2VL-7B | 44.4 | 58.1 | - | 63.7 |
| QuoTA | Qwen2-VL-2B+LLaVA-Video-7B | - | 71.9 | 59.0 | 65.9 |
| E-VRAG | ? | | 70.2 | 63.1 | 65.4 |
| APVR | LLM?+CLIP+Ground-DINO+Qwen2.5-VL-7B | - | - | 69.4 | 68.4 |
| Ours-GO | PerceptionEncoder+Qwen2.5-VL-7B | 52.7 | 72.3 | 63.4 | 66.7 |
| Ours-PO | PerceptionEncoder+Qwen2.5-VL-7B | 55.5 | 74.1 | 63.3 | 67.9 |

Table 9: Detailed results on VideoEvalPro-Open

| Method | LP | LR | HP | HR | Overall |
|---|---|---|---|---|---|
| LLaVA-7B | 28.5 | 13.6 | 20.7 | 19.3 | 24.2 |
| +Ours-GO | 40.8 | 19.7 | 20.7 | 22.3 | 32.7 |
| Qwen-7B | 33.9 | 15.6 | 24.8 | 17.8 | 27.7 |
| +Ours-GO | 43.3 | 18.4 | 30.6 | 22.3 | 35.0 |
| LLaVA-72B | 31.3 | 17.7 | 24.8 | 19.3 | 26.7 |
| +Ours-GO | 39.2 | 23.8 | 20.7 | 26.5 | 33.1 |
| Qwen-72B | 35.0 | 22.4 | 25.6 | 21.6 | 29.9 |
| +Ours-GO | 44.5 | 17.7 | 35.5 | 24.2 | 36.5 |

# B    DETAILED RESULTS ON VIDEOEVAL-PRO AND LVBENCH

## B.1    ADDITIONAL RESULTS ON VIDEOEVAL-PRO

We additionally report accuracy of 4 sub-tasks on both Open and MCQ subset: (1) **Local Perception (LP)**: Identify and retrieve visual elements or actions within short clips from long videos, covering object, action, attribute, and entity recognition, as well as segment and needle-in-a-haystack QA. (2) **Local Reasoning (LR)**: Perform reasoning over short event sequences, including temporal, causal, and object-action relationships within localized time windows. (3) **Holistic Perception (HP)**: Understand global visual patterns through aggregated spatial or structural information, primarily involving visual counting. (4) **Holistic Reasoning (HR)**: Achieve high-level understanding of long videos by reasoning about events, narrative structure, and underlying intent.

As shown in Table. 9 and Table. 10, Ours-GO achieves relatively higher accuracy gains on the local perception task. This improvement can be attributed to its retrieval-based architecture, which explicitly focuses on retrieving video segments most relevant to the question, thereby enhancing the model's sensitivity to local visual details.

Results in Table. 10 also show that Ours-PO achieves greater performance improvements on holistic reasoning task compared to Ours-GO. This is primarily because the option generation process in our framework is guided by local, question-relevant segments, which makes it challenging to infer answers that depend on global video semantics or require complex, multi-step reasoning over long-range dependencies. However, since Ours-PO leverages provided options—which ensure the inclusion of the correct answer—it mitigates this limitation and improving performance on holistic reasoning task.

## B.2    ADDITIONAL RESULTS ON LVBENCH

We additionally report accuracy on six sub-tasks: (1) **Entity Recognition (ER)**: Identify and track entities, their relations, actions, and associations over time. (2) **Event Understanding (EU)**: Rec-

Table 10: Detailed results on VideoEvalPro-MCQ

| Method | LP | LR | HP | HR | Overall |
|---|---|---|---|---|---|
| LLaVA-7B | 53.2 | 46.9 | 39.7 | 35.2 | 47.6 |
| +Ours-GO | 63.4 | 51.7 | 34.7 | 43.2 | 55.2 |
| +Ours-PO | 63.4 | 55.1 | 34.7 | 49.2 | 56.9 |
| Qwen-7B | 50.9 | 49.0 | 33.9 | 39.0 | 46.6 |
| +Ours-GO | 63.1 | 52.4 | 43.8 | 47.3 | 56.9 |
| +Ours-PO | 63.4 | 50.3 | 42.1 | 52.3 | 57.6 |
| LLaVA-72B | 54.6 | 57.1 | 32.2 | 41.7 | 50.1 |
| +Ours-GO | 63.4 | 62.6 | 33.1 | 53.0 | 58.3 |
| +Ours-PO | 64.5 | 61.9 | 35.5 | 58.0 | 60.1 |
| Qwen-72B | 60.5 | 59.9 | 38.0 | 48.9 | 55.9 |
| +Ours-GO | 67.8 | 59.2 | 47.9 | 53.0 | 61.9 |
| +Ours-PO | 70.1 | 60.5 | 46.3 | 57.2 | 64.2 |

Table 11: Detailed results on LVBench

| Method | ER | EU | KIR | TG | Rea | Sum | Overall |
|---|---|---|---|---|---|---|---|
| LLaVA-Video-7B | 43.6 | 40.0 | 39.5 | 34.5 | 46.8 | 32.8 | 42.0 |
| +Ours-GO | 54.8 | 49.1 | 55.7 | 35.0 | 44.8 | 32.8 | 51.3 |
| +Ours-PO | 57.8 | 51.3 | 59.5 | 47.3 | 48.8 | 36.2 | 54.2 |
| Qwen2.5-VL-7B | 44.3 | 43.9 | 50.5 | 40.9 | 50.2 | 36.2 | 45.5 |
| +Ours-GO | 53.5 | 49.8 | 62.2 | 41.8 | 54.2 | 36.2 | 52.7 |
| +Ours-PO | 57.3 | 52.9 | 62.5 | 41.4 | 53.2 | 34.5 | 55.5 |
| LLaVA-Video-72B | 44.6 | 45.0 | 48.8 | 39.5 | 50.7 | 37.9 | 46.1 |
| +Ours-GO | 55.5 | 49.0 | 58.4 | 37.7 | 49.3 | 32.8 | 51.7 |
| +Ours-PO | 59.2 | 52.7 | 60.5 | 45.5 | 52.2 | 34.5 | 54.8 |
| Qwen2.5-VL-72B | 49.2 | 49.1 | 54.0 | 36.8 | 56.7 | 34.5 | 49.6 |
| +Ours-GO | 55.4 | 49.9 | 63.6 | 37.3 | 54.7 | 41.4 | 54.0 |
| +Ours-PO | 58.6 | 54.1 | 65.6 | 42.7 | 55.7 | 31.0 | 56.9 |

ognize video-level semantics including genre, events, and scene changes. (3) **Key Information Retrieval (KIR)**: Extract precise factual details, such as on-screen text. (4) **Temporal Grounding (TG)**: Locate and describe events at specific timestamps. (5) **Reasoning (Rea)**: Perform causal, emotional, intentional, and prospective reasoning about video content. (6) **Summarization (Sum)**: Generate abstractive summaries capturing the full video narrative.

Consistent with the findings on VideoEval-Pro, the results in Table 11 demonstrate that Ours-GO achieves notable improvements on tasks that primarily require understanding of local visual content—such as entity recognition, event understanding, and key information retrieval. On more challenging, holistic tasks that demand comprehensive understanding and synthesis-such as reasoning and summarization, Ours-PO outperforms Ours-GO.

## C  QUALITATIVE COMPARISON

We provide several videos in the Supplementary Materials to visually compare our method with the baseline using the Qwen2.5-VL-7B backbone. The top of each video displays the given question and options. Below that, on the left side are the frames sampled by our method, and on the right side are the frames uniformly sampled by the baseline method. At the bottom, the similarity scores over video time in our method are visualized, such as Fig 4. Specifically, the horizontal axis denotes the video timeline (in seconds), where the blue curve represents our video-query-option similarity, the orange area indicates sampled frames (with darker shading corresponding to higher sampling density). The green line depicts similarity-based resolution allocation, while the red line represents uniform sampling resolution. The green scanline corresponds to our method, and the red scanline to

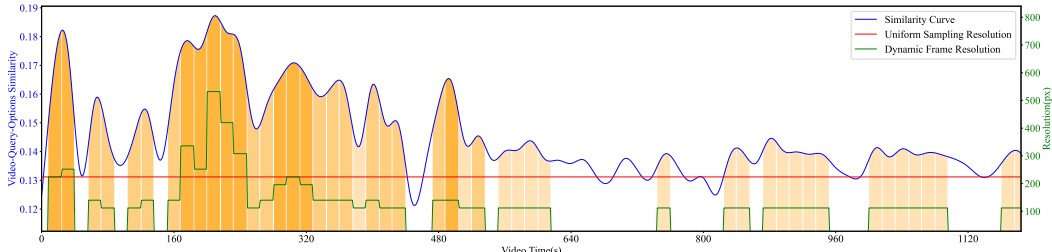

Figure 4: The horizontal axis denotes the video timeline (in seconds), where the blue curve represents video-query-option similarity, the orange area indicates sampled frames (with darker shading corresponding to higher sampling density). The green line depicts similarity-based resolution allocation, while the red line represents uniform sampling resolution.

Table 12: Benchmarks infomation

| Benchmark | Videos | QAs | Avg Duration |
|---|---|---|---|
| LVBench | 103 | 1549 | 67.3 min |
| VideoEvalPro | 465 | 1289 | 38.2 min |
| VideoMME | 900 | 2700 | 17.0 min |
| MLVU | 1122 | 2174 | 12.6 min |
| LongVideoBench | 735 | 1337 | 7.9 min |

uniform sampling. Along the right side of each scanline, the associated frame resolution is displayed. We provide two video examples:

(1) In the video "01262.mp4", to answer "What's the weakness of the villain?", our method—unlike uniform sampling—focuses sampling on the critical sequence where Tom and Jerry fetch water to confront the villain, guided by video-query-options similarity scores. At 2:42, the woman defeats the villain with a bucket of water, clearly revealing his weakness-water; at this moment, the similarity score peaks, sampling is densest, and resolution is highest (560×308), enabling Qwen2.5-VL-7B to correctly answer the question. Conversely, uniform sampling assigns all frames a small resolution (252x140) and includes many frames that are irrelevant to the question. As a result, uniform sampling approach leads to an incorrect answer.

(2) In the video "01254.mp4", when tasked with answering the question "How often do the people take water breaks?", our method strategically samples five key moments of people taking water breaks at a high resolution. At each of these moments, the time is clearly visible at the bottom of the screen. Leveraging this visible time information, the Qwen2.5-VL-7B can accurately infer that people take water breaks every 5 minutes. On the contrary, due to the relatively short duration of these five moments, uniform sampling fails to capture all of them. Additionally, because uniform sampling uses a low resolution, the Qwen2.5-VL-7B model is unable to recognize the time displayed on the screen. These factors ultimately lead to an incorrect answer when using uniform sampling.

## D   MORE EVALUTATION DETAILS

### D.1   BENCHMARKS DETAILS

In Table 12, we present the number of videos, the number of question - answer (QA) pairs, and the average video duration across the five long video benchmarks we utilized. Additionally, in Fig. 5, we illustrate the duration distribution of these five benchmarks. These demonstrate that LVBench and VideoEval-Pro contain a higher proportion of long videos compared to the other benchmarks. This characteristic enables our method to achieve better performance on these two benchmarks.

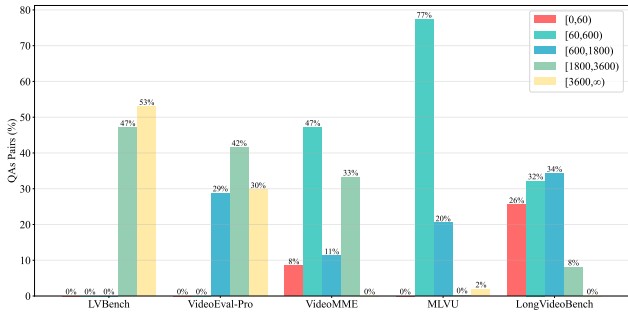

Figure 5: Benchmark duration distribution.

## D.2 EVALUATION PROMPTS

We provide the prompts used in our evalutaion on 5 long video benchmarks :

**LVBench**

```
{question}
(A) {optionA}
(B) {optionB}
......
Answer with the option's letter from the given choices directly.
```

**VideoEvalPro-MCQ**

```
Select the best answer to the following multiple-choice question based
on the video. Respond with only the letter (A, B, C, or D) of the
correct option, with no text around it.
{question} A. {optionA} B. {optionB} ......
```

**VideoEvalPro-Open**

```
{question} Keep the answer short and concise.
```

**VideoMME**

```
Select the best answer to the following multiple-choice question based
on the video and the subtitles. Respond with only the letter (A, B, C,
or D) of the correct option.
{question}
A. {optionA}
B. {optionB}
......
Answer with the option's letter from the given choices directly.
```

**MLVU**

```
Carefully watch this video and pay attention to every detail. Based on
your observations, select the best option that accurately addresses the
question.
Question: {question}
Options:
A. {optionA}
B. {optionB}
......
Only give the best option.
Best Option:
```

**LongVideoBench**

```
{question}
A. {optionA}
B. {optionB}
......
Answer with the option's letter from the given choices directly.
```

# E DETAILS OF OPTIONS GENERATION

## E.1 QUALITY METRICS DEFINITION

Formally, let $n$ denote the total number of evaluation questions. For the $i$-th question, let $Q_i$ be the question text, $A_i$ the ground-truth answer, and $O_i$ the set of generated options. We define SCOC (Semantic Correct Option Count) as the number of options in $O_i$ that are judged to be semantically equivalent to $A_i$, where semantic equivalence is determined via a GPT-based judgment. Specifically, we employ a simple prompt to instruct the large language model *gpt-4.1-2025-04-14* to count the number of semantically correct options for a given question, along with its corresponding answer and candidate options. The prompt used is as follows:

```
You are an intelligent chatbot designed to evaluate the correctness of
given options.
You will be provided with a question, a reference answer, and a set of
options.
Without relying on external knowledge, determine whether the correct
answer is included in the provided options, and count the number of
correct answers in the options.
```
Question: {question}
Reference Answer: {target}
Options: {gen_opts}
```
Please directly return the number of correct options, without any
additional text. If no option is correct, return 0.
```

The metrics **OCA** and **MPOC** are then computed as follows:

$$OCA = \frac{1}{n} \sum_{i=1}^{n} \mathbb{I}[SCOC_i > 0], \tag{10}$$

$$MPOC = \frac{1}{\sum_{i=1}^{n} \mathbb{I}[SCOC_i > 0]} \sum_{i=1}^{n} \left( \frac{SCOC_i}{|O_i|} \cdot \mathbb{I}[SCOC_i > 0] \right), \tag{11}$$

where $\mathbb{I}[\cdot]$ denotes the indicator function (equals 1 if the condition is true and 0 otherwise).

## E.2 DIFFERENT WAYS TO GENERATE OPTIONS

We compared options generated with and without using the video, and we also evaluated the impact of using stronger models (Seed1.5VL and GPT-5) for option generation (Table 13). All experiments were evaluated using the LLaVA-Video-7B model on the VideoEvalPro-MCQ dataset.

The results demonstrate that incorporating video context during option generation consistently enhances performance. In contrast, without video information, even powerful models like GPT-5 and Seed1.5VL yield only marginal gains over the baseline. Notably, when video context is used with a stronger model

Table 13: Different Ways to Generate Options

| Method | Option Model | With Video | Accuracy(%) |
|--------|-------------|------------|-------------|
| Base | - | - | 47.6 |
| Ours-PO | - | - | 56.9 |
| Ours-GO | LLaVA-Video-7B | Yes | 55.3 |
| Ours-GO | LLaVA-Video-7B | No | 53.9 |
| Ours-GO | Seed1.5VL | Yes | 56.2 |
| Ours-GO | Seed1.5VL | No | 54.9 |
| Ours-GO | gpt-5-2025-08-07 | No | 54.8 |

such as Seed1.5VL, performance improves substantially, approaching the accuracy of methods that are provided with options.

### E.3 CROSS-MODEL OPTION GENERATION

Cross-model option generation can yield similar—even improved performance. We evalute use different MLLMs to generate options on VideoEvalPro-MCQ benchmark:

The results in Table 14 suggest that our method is not overly dependent on the alignment between the option-generation and answering models, and cross-model option generation can still deliver strong, consistent improvements.

Table 14: Cross-model Option Generation

| Answering Model | Option Model | Accuracy(%) |
|---|---|---|
| LLaVA-Video-7B | - | 47.6 |
| LLaVA-Video-7B | LLaVA-Video-7B | 55.3 |
| LLaVA-Video-7B | Qwen2.5-VL-7B | 56.4 |
| Qwen2.5-VL-7B | - | 46.6 |
| Qwen2.5-VL-7B | LLaVA-Video-7B | 56.6 |
| Qwen2.5-VL-7B | Qwen2.5-VL-7B | 56.9 |

## F INCORPORATING GLOBAL INFORMATION

It is acknowledged that the proposed retrieval-based approach may potentially overlook certain global contextual cues inherent in the full video. To mitigate this limitation, we conducted extensive experiments to explore optimal strategies for integrating global context. Specifically, we investigated two existing strategies: "Retrieval By Global Summary Embedding" and "Context Fusion of Global Summary." Furthermore, we designed a novel approach, "Insert Global Frames," to facilitate the more effective incorporation of global information.

### F.1 RETRIEVAL BY GLOBAL SUMMARY EMEBDDING

We first generate a global textual summary of the entire video by uniformly sampling frames and prompting the MLLM with: "Please describe the video in detail." We then integrate this summary into the retrieval process in three ways:

(1) **APPEND**: The global summary is appended as an additional "statement" to the set in Equation 3 before computing the VQOS score (Equation 4). (2) **CONCAT**: The global summary is concatenated to each individual statement in Equation 3 before computing the VQOS score (Equation 4). (3) **MIX**: The global summary is encoded into an embedding and weighted-additively fused with the original text embeddings in Equation 4.

We evaluated these strategies using LLaVA-Video-7B on the VideoEvalPro-MCQ benchmark, the results in Table 15 show that: All three strategies resulted in performance degradation compared to our original method. Notably, as the weight $\lambda$ increases in the MIX variant, accuracy further declines. This suggests that the global textual summary often introduces irrelevant or misleading information that interferes with the relevance scoring process—particularly since the summary is generic and not question-aware.

Table 15: Results with global summary emebdding

| Method | Weight $\lambda$ | Accuracy(%) |
|---|---|---|
| Ours-PO | - | 56.9 |
| APPEND | - | 46.0 |
| CONCAT | - | 52.1 |
| MIX | 0.1 | 56.3 |
| MIX | 0.3 | 55.2 |
| MIX | 0.5 | 51.4 |

Table 16: Context fusion of global summary

| Method | Accuracy(%) w/o global | Accuracy(%) w/ global |
|---|---|---|
| Baseline | 47.6 | 43.6 |
| Ours-PO | 56.9 | 52.0 |

## F.2 CONTEXT FUSION OF GLOBAL SUMMARY

We also tested directly fusing the global summary into the input context by concatenating it with the original question (e.g., "[Question] + [Global summary]"). Results using LLaVA-Video-7B on VideoEvalPro-MCQ:

The results in Table 16 show that incorporating the generated global description consistently led to a performance drop in both settings. This indicates that the generated summary may introduce noisy or redundant information that interferes with downstream reasoning, rather than providing useful global context. One plausible explanation is that textual summaries are inherently lossy representations of the rich, multimodal information present in videos. Consequently, directly adding a global textual summary may be less effective than enhancing visual coverage through frame selection.

Table 17: Insert Global Frames

| T | Accuracy(%) |
|---|---|
| 0 | 56.9 |
| 4 | 56.9 |
| 8 | 56.8 |
| 16 | 57.4 |
| 32 | 55.9 |

## F.3 INSERT GLOBAL FRAMES

Motivated by these findings, we propose a visual-centric alternative: instead of relying on textual summaries, we enhance temporal coverage by explicitly inserting representative frames from under-represented segments of the video.

Specifically, after applying our original frame sampling method, we divide the long video into **T** uniform temporal segments. For any segment that contains no sampled frames, we add the frame at its temporal center. To maintain a fixed total number of frames (ensuring a fair comparison), we simultaneously remove the frame with the lowest VQOS score.

We evaluated this refined sampling strategy on LLaVA-Video-7B with varying values of T. The results in Table 17 show that moderate temporal segmentation (e.g., T=16 ) yields a slight but consistent performance gain, suggesting that strategically augmenting underrepresented temporal regions with representative frames can better capture global dynamics than relying on textual summaries. However, excessive segmentation (e.g., T=32 ) appears to disrupt the original sampling distribution, leading to a performance decline.

## G RUNTIME AND ACCELERATION

### G.1 ACCURACY AND RUNTIME TRADE-OFF

The main computational overhead of our method stems from computing the **VQOS**, which involves sampling frames from long videos at a specified **FPS** and extracting their embeddings. As discussed in Sec 4.4, this cost can be effectively reduced under limited computational budgets by either using a smaller **VTR** model, lowering the sampling FPS, or both.

To quantify this trade-off, we evaluate accuracy on LVBench and measure total runtime on the same single GPU setup, comparing our approach against baselines as well as other training-free methods: AKS and AdaReTake.

The results in Table 18 show that our method achieves a favorable balance between accuracy and efficiency. Even with a lightweight VTR model (e.g., PE-L/14) at a low sampling rate (0.5 FPS) yields significant accuracy gains over other methods while incurring only modest runtime overhead. In contrast, higher-fidelity configurations (e.g., PE-G/14 at 1.0 FPS) deliver peak performance—reaching 55.5% accuracy with Qwen2.5-VL-7B—but require substantially more computation.

Table 18: Accuracy and runtime trade-off

| Model | Method | VTR-Model | Retrieval FPS | Accuracy(%) | Runtime(hour) |
|-------|--------|-----------|---------------|-------------|---------------|
| LLaVA-Video | - | - | - | 42.0 | 2.4 |
| | AKS | - | 1.0 | 47.0 | 3.6 |
| | AdaReTake | - | - | 49.6 | 53.4 |
| | Ours-PO | PE-L/14 | 0.5 | 52.7 | 3.9 |
| | | | 1.0 | 52.0 | 5.5 |
| | | PE-G/14 | 1.0 | 54.2 | 29.0 |
| Qwen2.5-VL | - | - | - | 45.5 | 3.6 |
| | AKS | - | 1.0 | 45.0 | 4.8 |
| | AdaReTake | - | - | 51.0 | 35.9 |
| | Ours-PO | PE-L/14 | 0.5 | 54.5 | 5.2 |
| | | | 1.0 | 55.2 | 6.7 |
| | | PE-G/14 | 1.0 | 55.5 | 30.2 |

In practice, we can select the configuration that best aligns with their accuracy requirements and computational constraints. This flexibility makes our framework adaptable to diverse real-world deployment scenarios.

## G.2 ACCELERATION WITHOUT EXTRA MODEL

Notably, both the VTR model and the MLLM's visual encoder perform similar operations: extracting visual tokens and (in VTR's case) pooling them into video embeddings. This raises an important opportunity for optimization: *What if we eliminate redundant computation by reusing the visual tokens already extracted by the MLLM's visual encoder?* Specifically, instead of having VTR model computes frame embeddings and then recomputing visual tokens by MLLM, we could directly use the MLLM's native visual encoder to extract both the visual tokens and derive the pooled video segment embeddings. This would effectively eliminate time , since token extraction and embedding aggregation would occur in a single unified pass. Therefore, we could revise Algorithm 1 into Algorithm 2 by eliminating the additional VTR model.

For LLaVA-Video, the visual encoder is fine-tuned from SigLIP (Zhai et al., 2023), a model originally pretrained on large-scale image-text retrieval tasks — making it inherently well-suited for use as a VTR model. Notably, LLaVA-Video removes SigLIP's original vision pooling head (approximately 15M parameters), as it is unnecessary within the MLLM's architecture. In our implementation, we reintroduce the pooling head by directly copying it from the original SigLIP checkpoint, without any additional fine-tuning or parameter updates. Additionally, we directly leverage SigLIP's pretrained text encoder to compute text embeddings. Since the text encoder of SigLIP is lightweight and each question embedding is computed only once per query, the associated computational cost is negligible. While one could alternatively use the MLLM's native text encoder and attach a trainable pooling head to generate text embeddings, this would require additional fine-tuning — violating our design principle of training-free adaptation.

The resulting model, which we refer to as "SigLIP-LLaVA", is evaluated in Table 3. Remarkably, even without task-specific adaptation of the pooling head, SigLIP-LLaVA achieves strong performance on LVBench — outperforming the baseline by approximately 10%, and falling only 2.2% behind our best-performing model. This finding further supports the availability of Algorithm 2. In contrast, for Qwen2.5-VL, the visual encoder does not include a native pooling head suitable for generating video embeddings — meaning that introducing such a component would require additional training. Since the primary goal of this work is to explore training-free architectural optimizations, we do not implement this extension in our current pipeline. However, this remains a compelling avenue for future research.

## G.3 STREAMING LONG VIDEO UNDERSTANDING

Our method is primarily designed to enhance offline long video understanding capabilities without requiring any training, and as such, computational efficiency or inference latency is not the main

---

**Algorithm 2** Our Method Without Extra Model

---

**Require:** Video segments ($\mathcal{V}$), question ($q$), option generation round ($R$), sampled frame num ($N$), multimodal large language model (MLLM), Adaptive frame sampling (AFS), Dynamic Resolution Allocation (DRA).

**Ensure:** Answer the question according to the video.

1: $segment\_embeddings \leftarrow \emptyset$
2: $visual\_tokens\_cache \leftarrow \emptyset$
3: $simiarities \leftarrow \emptyset$
4: $generated\_options \leftarrow \emptyset$
5: $q\_embedding \leftarrow$ MLLM_text_encoder($q$)
6: **for** each $V_i \in \mathcal{V}$ **do**
7:    $visual\_tokens, visual\_embedding \leftarrow$ MLLM_visual_encoder($V_i$)
8:    $segment\_embedding$.append($visual\_embedding$)
9:    $visual\_tokens\_cache$.extend($visual\_tokens$)
10: **end for**
11: $S^0 \leftarrow cosine\_simiarity(q\_embedding, segment\_embeddings)$
12: $\mathcal{V}' \leftarrow TopK(\mathcal{V}, S^0, R \cdot N)$
13: **for** $r \leftarrow 1$ to $R$ **do**
14:    $\mathcal{V}_r \leftarrow (V_{r+kR})_{k=0}^{N-1}, \quad V \in \mathcal{V}'$
15:    $sampled\_frames \leftarrow$ DRA(AFS($S^0, \mathcal{V}_r$))
16:    $visual\_tokens \leftarrow visual\_tokens\_cache$.get($sampled\_frames$)
17:    $options \leftarrow$ MLLM($'Please \quad generate \quad some \quad options....'$, $visual\_tokens$)
18:    $generated\_options$.extend($options$)
19: **end for**
20: **for** each $o \in generated\_options$ **do**
21:    $o\_embedding \leftarrow$ MLLM_text_encoder($q + o$)
22:    $S \leftarrow cosine\_simiarity(o\_embedding, segment\_embeddings)$
23:    $similarities \leftarrow$ max($simiarities, S$)
24: **end for**
25: $\mathcal{V}_{final} \leftarrow TopK(\mathcal{V}, simiarities, N)$
26: $sampled\_frames \leftarrow$ DRA(AFS($simiarities, \mathcal{V}_{final}$))
27: $visual\_tokens \leftarrow visual\_tokens\_cache$.get($sampled\_frames$)
28: $answer \leftarrow$ MLLM($q, visual\_tokens$)
29: **return** $answer$

---

focus. Nevertheless, our approach can be easily adapted for streaming long-video understanding scenarios with minimal modifications.

As outlined in Algorithm 3, at certain sampling fps, the system checks whether the user has asked a question. If no question is detected, a frame is sampled from the video stream. The VTR model is used to compute the frame embedding, which is then cached. Once $T$ frames have been accumulated, the system computes a video segment embedding from these $T$ frame embeddings and stores it. If a question is posed, the VTR model is used to encode the question into an embedding. This question embedding is then compared with historical video segment embeddings to compute similarity scores (VQOS). Then, AFS and DRA are applied to sample $N$ relevant frames from the historical cache. These frames, along with the question, are fed into a Multimodal Large Language Model (MLLM) to generate an answer.

---

**Algorithm 3** Streaming Video Processing with Our Method

---

**Require:** Video stream, sampling fps ($fps$), video segment length ($T$), multimodal large language model (MLLM), video-query-options similairity (VQOS), adaptive frame sampling (AFS), dynamic resolution allocation (DRA).

**Ensure:** Answers to user questions on the streaming video.

1: $visual\_tokens\_cache \leftarrow \emptyset$
2: $segment\_embeddings \leftarrow \emptyset$
3: $frame\_embeds\_buffer \leftarrow \emptyset$
4: **while** video stream is active **do**
5:     **if** $video\_time \mod \frac{1}{fps} == 0$ **then**
6:         **if** user has question $q$ **then**
7:             $q\_embedding, text\_tokens \leftarrow$ MLLM_text_encoder$(q)$
8:             $similarities \leftarrow$ VQOS$(segment\_embeddings, q\_embedding)$
9:             $sampled\_visual\_tokens \leftarrow$ DRA(AFS$(similarities, visual\_tokens\_cache))$
10:           $answer \leftarrow$ MLLM$(text\_tokens, sampled\_visual\_tokens)$
11:           **Print** $answer$
12:         **else**
13:             $frame \leftarrow$ sample_frame$(video\_stream)$
14:             $frame\_embedding, visual\_tokens \leftarrow$ MLLM_visual_encoder$(frame)$
15:             $visual\_tokens\_cache$.append$(visual\_token)$
16:             $frame\_embeds\_buffer$.append$(frame\_embedding)$
17:           **if** len$(frame\_buffer) == T$ **then**
18:               $segment\_embedding \leftarrow$ aggregate$(frame\_buffer)$
19:               $segment\_embeddings$.append$(segment\_embedding)$
20:               $frame\_embeds\_buffer \leftarrow \emptyset$
21:           **end if**
22:         **end if**
23:     **end if**
24: **end while**

---

Notably, our method performs well under the configuration $fps = 1$ and $T = 16$, as demonstrated in Section 4. In streaming video scenarios, the time required for frame embedding extraction is negligible: even when using the largest video-text retrieval (VTR) model (PE-G/14), the frame embedding computation takes only $15.5/60 \approx 0.26$ seconds (Table 3), which is well below the inter-frame interval of $\frac{1}{fps} = 1$ second. Thus, embedding extraction introduces no bottleneck in streaming video processing.

