# OpenReview forum: "A Training-Free Framework for Long Video Understanding via Video-Query-Options Similarity"
_ICLR.cc/2026/Conference — ICLR 2026 Poster_

### Official Review · Reviewer_AQQ4 · 2025-10-21

**Soundness:** 2
**Presentation:** 2
**Contribution:** 2
**Rating:** 4
**Confidence:** 3

**Summary:**

This paper presents a training-free framework aimed at improving long video understanding in MLLMs by integrating Adaptive Frame Sampling (AFS), Dynamic Resolution Allocation (DRA), and Video-Query-Options Similarity (VQOS). The approach leverages a video-text retrieval backbone to guide query-conditioned frame sampling and resolution strategies. VQOS uses the MLLM to generate candidate answer options and fuses them with queries to perform more targeted retrieval, akin to a hypothesis-verification cognitive process. The system is implemented on LLaVA-Video and Qwen2.5-VL (both 7B and 72B sizes) and tested across five mainstream long video benchmarks, showing measurable improvements over SOTA training-free and training-based methods. Extensive ablation and component analyses are provided.

**Strengths:**

**1. Empirical Evidence:**
Table 1 (Page 6) and the associated ablations (Table 2, Table 3, Table 4, Table 6) demonstrate consistent and sizable gains on several long video understanding tasks, especially on LVBench and VideoEval-Pro. Improvements are especially marked for models applied to hour-long videos. Ablations quantify the effect of each component (AFS, DRA, option generation), while Figure 3 and Figure 4 provide insight into the trade-offs and behavior of the system—e.g., the diminishing returns in OCA versus MPCO as more option rounds are added (Page 9), as well as the visualization of similarity-guided sampling over time across the video timeline (Figure 4, Page 15).

**2. Generalization:**
 The model's robust generalization is further highlighted as the results show that method can be seamlessly integrated with various mainstream MLLM backbones (such as LLaVA-Video and Qwen2.5-VL), yielding consistent performance improvements across the board. This adaptability underscores its powerful generalization and transfer capabilities.

**Weaknesses:**

**1. Marginal originality:**
The core idea, using a retrieval model to select relevant video segments and adjusting sampling density, is essentially a combination of AKS (adaptive keyframe sampling) and AdaReTake (token-level redundancy reduction). The “Video-Query-Options Similarity” (VQOS) module is a minor twist: prompting the base MLLM to generate options and then fusing similarity across them. This is not a fundamental innovation but a straightforward heuristic that adds inference-time complexity.

**2. Option generation bias:**
The method lets the same MLLM that will later answer the question generate “candidate options.” This risks information leakage and circular reasoning—if the model already encodes the answer distribution, the retrieval pipeline simply re-weights what the model already “believes.” The ablations (Table 2) show small gains from generated options (+0.7%, +0.4%), suggesting this stage adds little actual reasoning power.

**3. VQOS dependence on retrieval quality.**
Performance gains (3–5%) largely stem from better retrieval models (PE-G/14). When using weaker CLIP backbones, the improvement vanishes. Thus, the method’s robustness across retrieval encoders or unseen domains is unproven.

**4. Figures are cluttered:**
Figure 2’s diagram is overly dense and still fails to clearly show the interaction between VQOS, AFS, and DRA. The legends mix fonts and arrows in ways that reduce readability.

**Questions:**

1. How do you ensure that this “option generation” does not leak answer information or overfit to the model’s own prior?
2. Would cross-model option generation (e.g., generate options with LLaVA, use Qwen2.5-VL for retrieval) yield similar improvements?

More questions please see the Weaknesses.

---

> ### Author Response · Authors · 2025-11-23
>
> **Thank you for your insightful feedback and appreciate your recognition of our work.**
> **Below, we provide detailed responses to your concerns:**
>
> ## Q1 : Marginal originality
>
> To clarify, our method is not a combination of AKS and AdaReTake.
>
> Compared to AKS, which selects frames based solely on the similarity between individual frames and the question, our approach operates at a video-segment level and leverages richer semantic information (as shown in Figure 1(b)). Specifically, we introduce the Video-Question-Options Similarity (VQOS), which jointly considers the question, candidate answer options, and video content to identify the most discriminative segments. Furthermore, within the selected high-scoring segments, we adaptively sample frames at higher rates and resize frames according to their VQOS scores (e.g., assigning higher resolution to more relevant frames). This dual-level adaptation—both in sampling density and visual fidelity—enhances the representation of critical information, a capability not present in AKS.
>
> As for AdaReTake, it is fundamentally different in objective and mechanism: AdaReTake is a token compression technique designed to reduce redundancy in visual tokens, thereby allowing more frames to be processed within a fixed context budget. Our method, by contrast, focuses on frame selection and representation. We only demonstrate in our experiments that AdaReTake can be seamlessly integrated with our pipeline to further boost performance, which highlights the compatibility and modularity of our framework.
>
>
> ## Q2: Option generation bias
>
> Indeed, using the same MLLM to generate answer options may be constrained by the model’s original output distribution. However, it is a fundamental limitation inherent to all training-free approaches. To mitigate this,  our method does not directly rely on the MLLM’s inherent answer-prediction capability. Instead, we prompt the model to hypothesize plausible candidate answers and then verify them through a retrieval-augmented reasoning process. Moreover, the average performance gains of +0.7% and +0.4% achieved by our generated options across five benchmarks are not marginal. For context, in AKS—where key frames are selected solely based on frame-question similarity—additional mechanisms beyond this basic retrieval yield only +0.3% on LongVideoBench and +0.9% on VideoMME.
>
> ## Q3: VQOS dependence on retrieval quality
>
> Table 3 shows that our method consistently improves performance across various VTR models, with larger gains achieved when using stronger (i.e., higher-performing) VTR models. Even with a weaker VTR model such as CLIP-B/32, our method achieves 49.8%, significantly outperforming the baseline of 42.0%. These results could demonstrate the robustness of our approach across different VTR backbones.
>
> ## Q4: Figures are cluttered
> For a better understanding of our method, please refer to Appendix D.
>
> ## Q5: Leak answer information
>
> Our option generation process does not leak ground-truth answer information, as the candidate options are generated solely by the original MLLM using only the video and question as input—without access to the true answer or any external supervision. Regarding the concern about “overfitting to the model’s own prior”: since our method is entirely training-free and involves no parameter updates, our method just harnesses the model's own prior , not overfitting it.
>
> ## Q6: Cross-model Option Generation
>
> Cross-model option generation can yield similar—even improved performance.  We evalute use different MLLMs to generate options on VideoEvalPro-MCQ benchmark:
>
> | Answering Model  | Option Model | Accuracy(%) |
> |------------------|---------------|-------------|
> |LLaVA-Video-7B |-              |47.6         |
> |LLaVA-Video-7B |LLaVA-Video-7B |55.3         |
> |LLaVA-Video-7B |Qwen2.5-VL-7B  |56.4         |
> |Qwen2.5-VL-7B  |-              |46.6         |
> |Qwen2.5-VL-7B  |LLaVA-Video-7B |56.6         |
> |Qwen2.5-VL-7B  |Qwen2.5-VL-7B  |56.9         |
>
>
> These results suggest that our method is not overly dependent on the alignment between the option-generation and answering models, and cross-model option generation can still deliver strong, consistent improvements.

---

### Official Review · Reviewer_1kjF · 2025-10-29

**Soundness:** 3
**Presentation:** 3
**Contribution:** 3
**Rating:** 6
**Confidence:** 4

**Summary:**

This paper presents a training-free framework for long video understanding that integrates three components—Adaptive Frame Sampling (AFS), Dynamic Resolution Allocation (DRA), and Video-Query-Options Similarity (VQOS)—to efficiently manage the token budget by focusing computational resources on relevant video segments, reportedly achieving state-of-the-art results on several benchmarks.

**Strengths:**

- The proposed framework is training-free and model-agnostic, making it a highly practical solution that can be readily applied to various off-the-shelf MLLMs (as demonstrated on LLaVA-Video and Qwen2.5-VL), enhancing its accessibility and potential for broad adoption.

**Weaknesses:**

1. I observed that the model's performance initially improves but then declines as the number of candidate options increases. Could the authors explain this non-monotonic behavior? Is this phenomenon related to the concept of pass@k?

​2. Could segmenting a long video into multiple short clips risk scattering information relevant to a long-term temporal question across different segments? If so, might this lead to incorrect segment localization when answering questions that require integrating evidence from across the entire video timeline?

**Questions:**

See Weaknesses

---

> ### Author Response · Authors · 2025-11-23
>
> **Thank you for your insightful feedback and appreciate your recognition of our work.**
> **Below, we provide detailed responses to your concerns:**
>
>
> ## Q1: Effect of Number of Candidate Options
>
> Figure 3 in our paper helps to address you question:
> Initially, generating more options improves performance because it raises the likelihood of including the correct answer—evidenced by the steady rise of OCA. When the correct option is present, the model’s final accuracy benefits accordingly. However, as the number of generated options continues to grow, the number of incorrect answers also  raises—reflected on the consistent decline of MPCO. This means that while the correct option is more likely to be included, it becomes increasingly diluted among a larger set of distracting or incorrect alternatives. This higher level of distraction ultimately undermines the model’s ability to select the right answer, leading to a drop in overall accuracy after a certain point. This explains the observed non-monotonic trend: performance first improves, reaches a peak, and then degrades with further increases in the number of candidate options.

---

> ### Author Response · Authors · 2025-11-23
>
> ## Q2:  Incorporating Global Information
>
> Thank you for this insightful and valuable suggestion! Indeed, our retrieval-based approach may inadvertently lose certain global contextual cues present in the full video.
>
> Motivated by your comment, we conducted a series of experiments to explore how best to integrate global context. Specifically, we investigated three strategies : "Retrieval By Global Summary Emebdding", "Context Fusion of Global Summary"—and further designed a novel approach "Insert Global Frames" to better incorporating global information.
>
> ###  1. Retrieval By Global Summary Emebdding
> We first generate a global textual summary of the entire video by uniformly sampling frames and prompting the MLLM with: “Please describe the video in detail.” We then integrate this summary into the retrieval process in three ways:
>
>  - **APPEND**: The global summary is appended as an additional “statement” to the set in Equation 3 before computing the VQOS score (Equation 4).
>  - **CONCAT**: The global summary is concatenated to each individual statement in Equation 3 before computing the VQOS score (Equation 4).
>  - **MIX**: The global summary is encoded into an embedding and weighted-additively fused with the original text embeddings in Equation 4.
>
> We evaluated these strategies using LLaVA-Video-7B on the VideoEvalPro-MCQ benchmark, with results as follows:
>
> | Method  | Weight λ |   Accuracy(%)|
> |---------|-------|--------------|
> | Ours-PO |-      | 56.9         |
> | APPEND  |-      | 46.0         |
> | CONCAT  |-      | 52.1         |
> | MIX    |0.1     | 56.3         |
> | MIX    |0.3     | 55.2         |
> | MIX    |0.5     | 51.4         |
>
>
> All three strategies resulted in performance degradation compared to our original method. Notably, as the weight λ increases in the MIX variant, accuracy further declines. This suggests that the global textual summary often introduces irrelevant or misleading information that interferes with the relevance scoring process—particularly since the summary is generic and not question-aware.
>
>
> ### 2. Context Fusion of Global Summary
>
> We also tested directly fusing the global summary into the input context by concatenating it with the original question (e.g., “[Question] + [Global summary]”).
> Results using LLaVA-Video-7B on VideoEvalPro-MCQ:
>
> | Method | Accuracy(%) w/o global| Accuracy(%) w/ global|
> |--------|------------------------|----------------------|
> |baseline|47.6                    |43.6                  |
> |Ours-PO |56.9                    |52.0                  |
>
>
> The results show that incorporating the generated global description consistently led to a performance drop in both settings. This indicates that the generated summary may introduce noisy or redundant information that interferes with downstream reasoning, rather than providing useful global context. One plausible explanation is that textual summaries are inherently lossy representations of the rich, multimodal information present in videos. Consequently, directly adding a global textual summary may be less effective than enhancing visual coverage through frame selection.
>
> ### 3. Insert Global Frames
>
> Motivated by these findings, we propose a visual-centric alternative: instead of relying on textual summaries, we enhance temporal coverage by explicitly inserting representative frames from underrepresented segments of the video.
>
> Specifically, after applying our original frame sampling method, we divide the long video into **T** uniform temporal segments. For any segment that contains no sampled frames, we add the frame at its temporal center. To maintain a fixed total number of frames (ensuring a fair comparison), we simultaneously remove the frame with the lowest VQOS score.
>
> We evaluated this refined sampling strategy on LLaVA-Video-7B with varying values of T :
>
> | T             | Accuracy(%) |
> |---------------|-------------|
> |0              |56.9         |
> |4              |56.9         |
> |8              |56.8         |
> |16             |57.4         |
> |32             |55.9         |
>
> The results show that moderate temporal segmentation (e.g., T=16 ) yields a slight but consistent performance gain, suggesting that strategically augmenting underrepresented temporal regions with representative frames can better capture global dynamics than relying on textual summaries. However, excessive segmentation (e.g., T=32 ) appears to disrupt the original sampling distribution, leading to a performance decline.
>
> Thank you for your good suggestion. We will continue to further explore and refine this research in the future.

---

### Official Review · Reviewer_9bhz · 2025-10-29

**Soundness:** 4
**Presentation:** 3
**Contribution:** 3
**Rating:** 8
**Confidence:** 4

**Summary:**

The paper proposes three techniques to improve video prefilling without fine-tuning the MLLM. Inspired by human cognition, the framework first generates multiple candidate answers and uses their text embeddings (via a PE-G encoder) to retrieve relevant video clips. Given a computational budget, it then applies adaptive frame and spatial-resolution sampling to prefill the selected clips into the MLLM. Experiments on five benchmarks demonstrate consistent improvements over the baseline by a clear margin.

**Strengths:**

1. The idea of generating candidate answers to guide retrieval is intuitive yet underexplored. Using answer-level text embeddings instead of query embeddings aligns well with how humans reason backward from possible outcomes, and the paper provides clear experimental validation of its effectiveness.


2. The ablations are well-structured and provide good insight into how each component contributes to the overall improvement.

**Weaknesses:**

1. The proposed two-pass pipeline introduces additional computation compared to single-pass or efficient prefilling methods. It would be helpful if the paper discussed the runtime overhead and its trade-off with accuracy.

2. Since only the retrieved clips are passed to the MLLM, global contextual information might be lost. Have the authors considered combining local retrieval with a global summary embedding or a context fusion?

**Questions:**

It would be interesting to understand how the quality of the generated answer options affects final performance. For example, what happens if the options are produced by an MLLM without seeing the video (i.e., based purely on a textual question)? In addition, would using a stronger model (e.g., GPT-5) to generate the options lead to further gains?

---

> ### Author Response · Authors · 2025-11-23
>
> ## Q1: Accuracy and Runtime Trade-off
>
> The main computational overhead of our method stems from computing the **VQOS**, which involves sampling frames from long videos at a specified **FPS** and extracting their embeddings. As discussed in Section 4.4, this cost can be effectively reduced under limited computational budgets by either using a smaller **VTR** model, lowering the sampling FPS, or both.
>
> To quantify this trade-off, we evaluate accuracy on LVBench and measure total runtime on the same single H20 GPU setup, comparing our approach against baselines as well as other training-free methods: AKS and AdaReTake.
>
> Results with LLaVA-Video-7B:
> | Method  | VTR-Model  | Retrieval FPS |Accuracy(%)|Runtime(hour)|
> |---------|------------|-----|-----------|-------------|
> |Baseline        |-           |-    |42.0       |2.4          |
> |AKS      |-           |1.0    |47.0       |3.6          |
> |AdaReTake|-           |-    |49.6       |53.4         |
> |Ours-PO  |PE-L/14     |0.5  |52.7       |3.9          |
> |Ours-PO  |PE-L/14     |1.0  |52.0       |5.5          |
> |Ours-PO  |PE-G/14     |1.0  |54.2       |29.0         |
>
> Results with Qwen2.5-VL-7B:
> | Method  | VTR-Model  | Retrieval FPS |Accuracy(%)|Runtime(hour)|
> |---------|------------|-----|-----------|-------------|
> |Baseline        |-           |-    |45.5       |3.6          |
> |AKS      |-           |1.0    |45.0       |4.8          |
> |AdaReTake|-           |-    |51.0       |35.9         |
> |Ours-PO  |PE-L/14     |0.5  |54.5       |5.2          |
> |Ours-PO  |PE-L/14     |1.0  |55.2       |6.7          |
> |Ours-PO  |PE-G/14     |1.0  |55.5       |30.2         |
>
> The results show that our method achieves a favorable balance between accuracy and efficiency. Even with a lightweight VTR model (e.g., PE-L/14) at a low sampling rate (0.5 FPS) yields significant accuracy gains over other methods while incurring only modest runtime overhead. In contrast, higher-fidelity configurations (e.g., PE-G/14 at 1.0 FPS) deliver peak performance—reaching 55.5% accuracy with Qwen2.5-VL-7B—but require substantially more computation.

---

> ### Author Response · Authors · 2025-11-23
>
> ## Q2:  Incorporating Global Information
>
> Thank you for this insightful and valuable suggestion! Indeed, our retrieval-based approach may inadvertently lose certain global contextual cues present in the full video.
>
> Motivated by your comment, we conducted a series of experiments to explore how best to integrate global context. Specifically, we investigated two strategies aligned with your suggestion—"Retrieval By Global Summary Emebdding" and "Context Fusion of Global Summary"—and further designed a novel approach "Insert Global Frames" to better incorporating global information.
>
> ###  1. Retrieval By Global Summary Emebdding
> We first generate a global textual summary of the entire video by uniformly sampling frames and prompting the MLLM with: “Please describe the video in detail.” We then integrate this summary into the retrieval process in three ways:
>
>  - **APPEND**: The global summary is appended as an additional “statement” to the set in Equation 3 before computing the VQOS score (Equation 4).
>  - **CONCAT**: The global summary is concatenated to each individual statement in Equation 3 before computing the VQOS score (Equation 4).
>  - **MIX**: The global summary is encoded into an embedding and weighted-additively fused with the original text embeddings in Equation 4.
>
> We evaluated these strategies using LLaVA-Video-7B on the VideoEvalPro-MCQ benchmark, with results as follows:
>
> | Method  | Weight λ |   Accuracy(%)|
> |---------|-------|--------------|
> | Ours-PO |-      | 56.9         |
> | APPEND  |-      | 46.0         |
> | CONCAT  |-      | 52.1         |
> | MIX    |0.1     | 56.3         |
> | MIX    |0.3     | 55.2         |
> | MIX    |0.5     | 51.4         |
>
>
> All three strategies resulted in performance degradation compared to our original method. Notably, as the weight λ increases in the MIX variant, accuracy further declines. This suggests that the global textual summary often introduces irrelevant or misleading information that interferes with the relevance scoring process—particularly since the summary is generic and not question-aware.
>
>
> ### 2. Context Fusion of Global Summary
>
> We also tested directly fusing the global summary into the input context by concatenating it with the original question (e.g., “[Question] + [Global summary]”).
> Results using LLaVA-Video-7B on VideoEvalPro-MCQ:
>
> | Method | Accuracy(%) w/o global| Accuracy(%) w/ global|
> |--------|------------------------|----------------------|
> |baseline|47.6                    |43.6                  |
> |Ours-PO |56.9                    |52.0                  |
>
>
> The results show that incorporating the generated global description consistently led to a performance drop in both settings. This indicates that the generated summary may introduce noisy or redundant information that interferes with downstream reasoning, rather than providing useful global context. One plausible explanation is that textual summaries are inherently lossy representations of the rich, multimodal information present in videos. Consequently, directly adding a global textual summary may be less effective than enhancing visual coverage through frame selection.
>
> ### 3. Insert Global Frames
>
> Motivated by these findings, we propose a visual-centric alternative: instead of relying on textual summaries, we enhance temporal coverage by explicitly inserting representative frames from underrepresented segments of the video.
>
> Specifically, after applying our original frame sampling method, we divide the long video into **T** uniform temporal segments. For any segment that contains no sampled frames, we add the frame at its temporal center. To maintain a fixed total number of frames (ensuring a fair comparison), we simultaneously remove the frame with the lowest VQOS score.
>
> We evaluated this refined sampling strategy on LLaVA-Video-7B with varying values of T :
>
> | T             | Accuracy(%) |
> |---------------|-------------|
> |0              |56.9         |
> |4              |56.9         |
> |8              |56.8         |
> |16             |57.4         |
> |32             |55.9         |
>
> The results show that moderate temporal segmentation (e.g., T=16 ) yields a slight but consistent performance gain, suggesting that strategically augmenting underrepresented temporal regions with representative frames can better capture global dynamics than relying on textual summaries. However, excessive segmentation (e.g., T=32 ) appears to disrupt the original sampling distribution, leading to a performance decline.
>
> Thank you for your good suggestion. We will continue to further explore and refine this research in the future.

---

> ### Author Response · Authors · 2025-11-23
>
> ## Q3: Different Ways to Generate Options
>
>
> We compared options generated with and without using the video, and we also evaluated the impact of using stronger models (Seed1.5VL and GPT-5) for option generation. All experiments were evaluated using the LLaVA-Video-7B model on the VideoEvalPro-MCQ dataset.
>
> |Method |Option Model    | Generate Options with Video | Accuracy(%) |
> |-------|----------------|-----------------------------|-------------|
> |-      |-               |-                            |47.6         |
> |Ours-PO|-               |-                            |56.9         |
> |Ours-GO|LLaVA-Video-7B  |✓                            |55.3         |
> |Ours-GO|LLaVA-Video-7B  |✗                            |53.9         |
> |Ours-GO|Seed1.5VL       |✓                            |56.2         |
> |Ours-GO|Seed1.5VL       |✗                            |54.9         |
> |Ours-GO|gpt-5-2025-08-07|✗                            |54.8         |
>
>
> The results demonstrate that incorporating video context during option generation consistently enhances performance. In contrast, without video information, even powerful models like GPT-5 and Seed1.5VL yield only marginal gains over the baseline. Notably, when video context is used with a stronger model such as Seed1.5VL, performance improves substantially, approaching the accuracy of methods that are provided with options.

---

### Meta-Review · Area_Chair_Bwqp · 2026-01-03

**Summary:**

The rebuttal solved most concerns from the reviewers. After reading the paper and rebuttal, the AC tends to accept this paper

**Reviewer Concerns:**

Most concerns are solved

**Reviewer Scores:**

Both reviewers 9bhz and 1kjF give a positive rating to the paper.
For the reviewer AQQ4, after reading the review and the comments from the authors, the AC agrees with the authors that the review is AI-generated with poor quality.

---

### Decision · Program_Chairs · 2026-01-26

Accept (Poster)